# On Time-Dependent Rheology of Sutterby Nanofluid Transport across a Rotating Cone with Anisotropic Slip Constraints and Bioconvection

**DOI:** 10.3390/nano12172902

**Published:** 2022-08-24

**Authors:** Sohaib Abdal, Imran Siddique, Khadijah M. Abualnaja, Saima Afzal, Mohammed M. M. Jaradat, Zead Mustafa, Hafiz Muhammad Ali

**Affiliations:** 1School of Mathematics, Northwest University, No.229 North Taibai Avenue, Xi’an 710069, China; 2Department of Mathematics, Khawaja Fareed University of Engineering and Information Technology, Rahim Yar Khan 64200, Pakistan; 3Department of Mathematics, University of Management and Technology, Lahore 54770, Pakistan; 4Department of Mathematics and Statistics, College of Science, Taif University, P.O. Box 11099, Taif 21944, Saudi Arabia; 5Mathematics Program, Department of Mathematics, Statstics and Physics, College of Arts and Sciences, Qatar University, Doha 2713, Qatar; 6Mechanical Engineering Department, King Fahd University of Petroleum & Minerals, Dhahran 31261, Saudi Arabia; 7Interdisciplinary Research Center for Renewable Energy and Power Systems (IRC-REPS), King Fahd University of Petroleum and Minerals, Dhahran 31261, Saudi Arabia

**Keywords:** Sutterby fluid, nanofluid, bioconvection, anisotropic slips, rotating cone, Runge–Kutta scheme

## Abstract

The purpose and novelty of our study include the scrutinization of the unsteady flow and heat characteristics of the unsteady Sutterby nano-fluid flow across an elongated cone using slip boundary conditions. The bioconvection of gyrotactic micro-organisms, Cattaneo–Christov, and thermal radiative fluxes with magnetic fields are significant physical aspects of the study. Anisotropic constraints on the cone surface are taken into account. The leading formulation is transmuted into ordinary differential formate via similarity functions. Five coupled equations with nonlinear terms are resolved numerically through the utilization of a MATLAB code for the Runge–Kutta procedure. The parameters of buoyancy ratio, the porosity of medium, and bioconvection Rayleigh number decrease x-direction velocity. The slip parameter retard y-direction velocity. The temperature for Sutterby fluids is at a hotter level, but its velocity is vividly slower compared to those of nanofluids. The temperature profile improves directly with thermophoresis, v-velocity slip, and random motion of nanoentities.

## 1. Introduction

In the modern era, many researchers worked on fluid flows that pass through a cone because of its progress in advanced technologies. It have many outstanding applications in industrial and engineering fields such as aeronautical engineering, electronic chips, endoscopy scanning, etc. The effect of chemical reaction on Casson fluid by using cone geometry was analyzed by Deebani et al. [1]. Verma et al. [2] numerically discussed the effects of Soret and Dufour with thermal radiation on MHD flow around a vertical cone. The two-dimensional MHD nanofluid flow passing over a plate or cone was discussed by Ahmad et al. [3]. The investigations of MHD micropolar fluid in the presence of porous medium passing across a cone were studied by Ahmad et al. [4]. Hazarika et al. [5] discussed the effect of variable viscosity of time-dependent micropolar fluid passing over a vertical cone. Dawar et al. [6] used non-isothermal and non-iso-solutal boundary conditions for Williamson nanofluid flow passing through two geometries. Nabwey and Mahdy [7] discussed the impact of non-linear temperature on micropolar fluid flow passing across the cone.

The basic premise of MHD is simple: An electrical conductor fluid, such as seawater, is used to create a unidirectional current. Naseem et al. [8] proposed an analytical treatment for MHD non-Newtonian micro liquid caused by plate stretching in the presence of Brownian motion, concluding that the thermal relaxation parameter has the potential to increase fluid temperature. The magneto hydrodynamic free convective boundary layer flow of a chemically reacting nanofluid from a convectively heated permeable vertical surface was investigated by Uddin et al. [9]. Waqas et al. [10] described a magneto hydrodynamic (MHD) mixed convection flow of micropolar liquid caused by a nonlinear stretched sheet under convective conditions. Srinivas et al. [11] investigated the effect of a chemical reaction on the MHD flow of a nanofluid in a porous pipe that was expanding or contracting in the presence of a heat source/sink. Habib et al. [12] analyzed numerical simulations for slip impacts on MHD nanofluid in the presence of an electromagnetic field and gyrotactic microbes under the influences of activation energy and heat radiation. Refs. [13,14,15,16] scrutinized the aspects of MHD flow of nanofluids.

Choi [17] used the term “nanofluid” in 1995 to describe a novel kind of heat transfer fluid that relies on suspending Nanoscale particles of metallic origin with an average particle size of less than 100 nm inside traditional heat transfer fluids. Nanofluids have been discovered to have improved thermo-physical properties such as thermal conductivity, thermal diffusivity, viscosity, and convective heat transfer coefficients when compared to base fluids such as oil or water in earlier studies. Khan et al. [18] investigated the magnetohydrodynamic thin layer nanofluid applied on a stretching cylinder including heat transfer. Kuznetsov and Nield [19] recently looked at the boundary layer in Newtonian and porous media filled with nanofluid over constant and movable boundaries. Turkyilmazoglu [20] investigated the flow of nanofluid boundary layers across a revolving disc. Hsiao [21] used an applied thermal system for micropolar nanofluid flow past over a stretching sheet. Some other researchers [22,23,24] explored the features of nanofluids.

Bioconvection occurs when the self-propelled kinetic microorganisms increase the density of the normal liquid in a specific direction to a specific direction, where the density gradually moves upward in the moving fluid in the normal fluid. Nayake et al. [25] investigated the velocity of the Casson flow of the electromagnetic sheet exponentially developing when the nanofluid incurs chemical, thermal, isolated, and moving microbial slip effects. Mansour et al. [26] explained bioconvection of magneto-hydrodynamic in the closed-loop cave. Balla et al. [27] analyzed the activity of living microorganisms in the porous square hole. Khan et al. [28] analyzed the bioconvection nanofluidic flow along with the extended former Oldroyd-B walls. Khan et al. [29] discussed the flow of a viscoelastic nanofluid across a moving surface, which is porous, along with the occurrence of microorganisms. Ali et al. [30] investigated the influence of Stefan blowing on Cattaneo–Christov attributes and the bioconvection of self-motivated microorganisms blending in water-based nanoparticles with a leading-edge ablation/accretion.

To the best of the researcher’s knowledge, no analysis of the unsteady Sutterby nano-fluid, heat, and mass characteristics of ordinary nanofluid flow across an elongated cone, keeping in view the conical boundaries in various technological industrial applications, has been performed. Latiff et al. [31] examined theoretically and arithmetically the MHD bioconvective of nanofluid flow near a spinning cone with anisotropic velocity slips, heat slips, density slips, and microbes slips. However, they did not contemplate unsteady Sutterby nanofluid and, due to sedimentation of nanofluids, we use living micro-organisms. The target and novelty of our latest study is to examine the unsteady flow and heat characteristics of the unsteady Sutterby nano-fluid flow across an elongated cone using slip boundary conditions. The physical aspects are improved with bioconvection, applied magnetic force, Cattaneo–Christov diffusion, and anisotropic constraints. The differentiated results are achieved numerically and discussed in detail.

## 2. Physical Model and Mathematical Formulation

This study contains axisymmetric, unsteady Sutterby nanofluid flow across a rotating cone with self motive micro-organisms. Moreover, rectangular curvilinear coordinate structure is assumed to be stable. The presence of buoyancy forces that are present in the flow depends on mass, temperature, and micro-organism difference. $u_{x}$, $v_{y}$, and $w_{z}$ represent velocity components along x, y and z-axis. B (magnetic field of strength) is normal to the rotating cone. The graphical representation of the physical formation is revealed in Figure 1. Here, $T_{w}$, *T*, *C*, *n*$C_{w}$, and $n_{w}$ represents fluid temperature at the wall, fluid temperature, nanoparticle volume fraction, and motile micro-organism density. Nanoparticle volume fraction and motile micro-organism density at the wall and $T_{\infty }$, $C_{\infty }$, and $n_{\infty }$ are taken away from the wall. With these assumptions, the momentum and mass equations in x and z directions, energy, concentration, and micro-organism conservation, which depends on time, are given below:
(1)$$\begin{array}{c} \partial_{x}u_{x}+\partial_{z}w_{z}+\frac{u_{x}}{x}=0, \end{array}$$
(2)$$\begin{array}{c} \partial_{t}u_{x}+u_{x}\partial_{x}u_{x}+w_{z}\partial_{z}u_{x}-\frac{v_{y}^{2}}{x}=\frac{\nu }{2\rho }\partial_{zz}u_{x}\left[1-\frac{Sb^{2}}{2}{\left(\partial_{z}u_{x}\right)}^{2}\right]-\frac{\sigma B^{2}u_{x}}{\rho }\\ +\frac{1}{\rho }\left[\left(1-C_{\infty }\right)\rho \beta g\left(T-T_{\infty }\right)-\left(\rho_{p}-\rho \right)g\left(C-C_{\infty }\right)-\left(n-n_{\infty }\right)g\gamma \left(\rho_{m}-\rho \right)\right], \end{array}$$
(3)$$\begin{array}{c} \partial_{t}v_{y}+u_{x}\partial_{x}v_{y}+w_{z}\partial_{z}v_{y}-\frac{v_{y}u_{x}}{x}=\frac{\nu }{2\rho }\partial_{zz}v_{y}\left[1-\frac{Sb^{2}}{2}{\left(\partial_{z}v_{y}\right)}^{2}\right]-\frac{\sigma B^{2}v_{y}}{\rho }, \end{array}$$
(4)$$\begin{array}{c} \partial_{t}T+u_{x}\partial_{x}T+w_{z}\partial_{z}T=\alpha \partial_{zz}T+\tau D_{B}\partial_{z}T\partial_{z}C+\tau \frac{D_{T}}{T_{\infty }}{\left(\partial_{z}T\right)}^{2}, \end{array}$$
(5)$$\begin{array}{c} \partial_{t}C+u_{x}\partial_{x}C+w_{z}\partial_{z}C=D_{B}\partial_{zz}C+\frac{D_{T}}{T_{\infty }}\partial_{zz}T, \end{array}$$
(6)$$\begin{array}{c} \partial_{t}n+u_{x}\partial_{x}n+w_{y}\partial_{z}n+\frac{c_{b}W_{c}}{C_{w}-C_{\infty }}\left[\partial_{z}\left(n\partial_{z}C\right)\right]=D_{m}\partial_{zz}n. \end{array}$$

Here, *t* represents time, fluid density is ρ, gravity acceleration is *g*, fluid viscosity is μ, deportment index of flow is *S*, consistency index is $b^{2}$, thermal diffusivity is α, the thermal expansion coefficient of the base fluid is β, the average volume of micro-organism is γ, micro-organism density is $\rho_{m}$, the ratio of heat capacity of nanofluid to the base fluid is τ, Brownian motion is $D_{B}$, the thermophorsis coefficient is $D_{T}$, the chemotaxis constant is $c_{b}$, the swimming speed of cell is Wc, and the micro-organism diffusivity coefficient is $D_{m}$.

Boundary conditions along with slip conditions are taken into account [32,33]:(7)$$\begin{array}{c} \left.\begin{array}{c} u_{x}=U_{1}\left(x,t\right)\nu \partial_{z}u_{x},\;v_{y}=V_{1}\left(x,t\right)\nu \partial_{z}v_{y}+x\left(\Omega sin\alpha \right){\left[1-s\left(\Omega sin\alpha \right)t\right]}^{-1},\;w_{z}=0,\\ T=T_{w}\left(x,t\right)+T_{1}\left(x,t\right)\partial_{z}T,C=C_{w}\left(x,t\right)+\;C_{1}\left(x,t\right)\partial_{z}C,\;n=n_{w}\left(x,t\right)+n_{1}\left(x,t\right)\partial_{z}n,\;at\;y=0,\\ u_{x}\to 0,\;v_{y}\to 0,\;T\to T_{\infty },\;C\to C_{\infty },\;n\to n_{\infty }\;as\;y\to \infty . \end{array}\right\} \end{array}$$
where dimensionless angular velocity of the cone is Ω. $U_{1}$, $V_{1}$, $T_{1}$, $C_{1}$, and $n_{1}$ represent velocity, temperature, concentration, and micro-organism density slips.

Introduce the following similarity transformation variables to proceed with the investigation [33].
(8)$$\begin{array}{c} \left.\begin{array}{c} \eta =\sqrt{\frac{\Omega sin\alpha }{\nu (1-st)}}z,\;u_{x}=\frac{-x\Omega sin\alpha }{2(1-st)}f^{\prime }\left(\eta \right),\;v_{y}=\frac{x\Omega sin\alpha }{\left(1-st\right)}g\left(\eta \right),\;w_{z}=\sqrt{\frac{\nu \Omega sin\alpha }{\left(1-st\right)}}f\left(\eta \right),\\ T=T_{\infty }+\left(T_{w}-T_{\infty }\right)\theta \left(\eta \right),\;T_{w}-T_{\infty }=\left[{\left(T_{w}\right)}_{0}-T_{\infty }\right]\frac{x}{L}{\left(1-st\right)}^{-2},\\ C=C_{\infty }+\left(C_{w}-C_{\infty }\right)\phi \left(\eta \right),\;C_{w}-C_{\infty }=\left[{\left(C_{w}\right)}_{0}-C_{\infty }\right]\frac{x}{L}{\left(1-st\right)}^{-2},\\ n=n_{w}\chi \left(\eta \right),\;n_{w}={\left(n_{w}\right)}_{0}\frac{x}{L}{\left(1-st\right)}^{-2},\;B^{2}\left(x,t\right)=B_{0}^{2}sin\alpha {\left(1-st\right)}^{-1},\;t=\left(\Omega sin\alpha \right)t_{1}. \end{array}\right\} \end{array}$$

Equation (Equation 1) is satisfied by using Equation (Equation 8), and Equations (2)–(6) are reduced into following differential equations.
(9)$$\begin{array}{c} \left[1-\frac{S}{8}ReDef^{\prime \prime 2}\right]f^{\prime \prime \prime }+\frac{1}{2}f^{\prime 2}-ff^{\prime \prime }-2g^{2}-2A\left(f^{\prime }+\frac{1}{2}\eta f^{\prime \prime }\right)-Mf^{\prime }-2\lambda \left(\theta -Nr\phi -Rb\chi \right)=0, \end{array}$$
(10)$$\begin{array}{c} {}[1-\frac{S}{2}ReDeg^{\prime 2}]g^{\prime \prime }-2fg^{\prime }+f^{\prime }g-2A\left(g+\frac{1}{2}\eta g^{\prime }\right)-Mg=0, \end{array}$$
(11)$$\begin{array}{c} \theta^{\prime \prime }-Pr\left[f\theta^{\prime }-\frac{1}{2}f^{\prime }\theta +A\left(2\theta +\frac{1}{2}\eta \theta^{\prime }\right)\right]+Nb\theta^{\prime }\phi^{\prime }+Nt\theta^{\prime 2}=0, \end{array}$$
(12)$$\begin{array}{c} \phi^{\prime \prime }-Sc\left[f\phi^{\prime }-\frac{1}{2}f^{\prime }\phi +A\left(2\phi +\frac{1}{2}\eta \phi^{\prime }\right)\right]+\left(\frac{Nb}{Nt}\right)\theta^{\prime \prime }=0, \end{array}$$
(13)$$\begin{array}{c} \chi^{\prime \prime }-Lb\left[f\chi^{\prime }-\frac{1}{2}f^{\prime }\chi +A\left(2\chi +\frac{1}{2}\eta \chi^{\prime }\right)\right]-Pe\left[\phi^{\prime \prime }+\chi^{\prime }\phi^{\prime }\right]=0. \end{array}$$
(14)$$\begin{array}{c} \left.\begin{array}{c} f\left(0\right)=S,\;f^{\prime }\left(0\right)=\Gamma_{u}f^{\prime \prime }\left(0\right),\;g\left(0\right)=1+\Gamma_{v}g^{\prime }\left(0\right),\;\theta \left(0\right)=1+\Gamma_{T}\theta^{\prime }\left(0\right),\\ \phi \left(0\right)=1+\Gamma_{C}\phi^{\prime }\left(0\right),\;\chi \left(0\right)=1+\Gamma_{n}\chi^{\prime }\left(0\right),\;at\;\eta =0,\\ f^{\prime }\left(\infty \right)\to 0,\;g\left(\infty \right)\to 0,\;\theta \left(\infty \right)\to 0,\phi \left(\infty \right)\to 0,\chi \left(\infty \right)\to 0,\;as\;\eta \to \infty . \end{array}\right\} \end{array}$$

The non-dimensional parameters in their respective orders are given below:

$Re=\frac{x^{2}\Omega sin\alpha }{\nu (1-st)}$ is the Sutterby Reynolds number, $De={\left(\frac{b\Omega sin\alpha }{\left(1-st\right)}\right)}^{2}$ is the Sutterby Deborah number, $Nr=\frac{\left(\rho_{p}-\rho \right)\left(C_{w}-C_{\infty }\right)}{\beta \rho \left(1-C_{\infty }\right)\left(T_{w}-T_{\infty }\right)}$ is the buoyancy ratio, $Rb=\frac{\left(\rho_{m}-\rho \right)\gamma n_{\infty }}{\left(1-C_{\infty }\right)\rho \beta \left(T_{w}-T_{\infty }\right)}$ is the bio-convection Rayleigh number, the Prandtl number is $Pr=\frac{\nu }{\alpha }$, the mixed convection parameter is $\lambda =\frac{Gr}{Re_{L}^{2}}$, $Gr=g\beta cos\alpha \left(1-C_{\infty }\right)\left(T_{w}-T_{\infty }\right)\frac{L^{3}}{\nu^{2}}$ is the Grashoff number, $Re_{L}^{2}=\Omega sin\alpha \frac{L^{2}}{\nu }$ is the Rayleigh number, the magnetic field is $M=\frac{\sigma B_{0}^{2}}{\rho \Omega }$, $Nb=\frac{\tau D_{B}\left(C_{w}-C_{\infty }\right)}{\nu }$ denotes Brownian motion, $Nt=\frac{\tau D_{T}\left(T_{w}-T_{\infty }\right)}{\nu T_{\infty }}$ is the thermophoresis parameter, the bioconvection Lewis number is $Lb=\frac{\alpha }{D_{m}}$, the bioconvection Peclet number is $Pe=\frac{bWc}{D_{m}}$, $Sc=\frac{\nu }{D_{B}}$ is the Schmidt number, $\Gamma_{u}=U_{1}\sqrt{\frac{\nu \Omega sin\alpha }{\left(1-st\right)}}$ is the u-velocity slip, $\Gamma_{v}=V_{1}\sqrt{\frac{\Omega sin\alpha }{\nu (1-st)}}$ is the v-velocity slip, $\Gamma_{T}=T_{1}\sqrt{\frac{\Omega sin\alpha }{\nu (1-st)}}$ is the thermal slip, $\Gamma_{C}=C_{1}\sqrt{\frac{\Omega sin\alpha }{\nu (1-st)}}$ is the solutal slip, and $\Gamma_{n}=n_{1}\sqrt{\frac{\Omega sin\alpha }{\nu (1-st)}}$ is motile density slip.

The physical quantities are stated as follows.

$C_{fx}$, $C_{fy}$, $Nu_{x}$, $Sh_{x}$, and $Nn_{x}$ are given below.

$C_{fx}=\frac{\tau_{xz}}{\rho {\left(\Omega xsin\alpha {\left(1-st\right)}^{-1}\right)}^{2}}$, $C_{fy}=\frac{\tau_{yz}}{\rho {\left(\Omega xsin\alpha {\left(1-st\right)}^{-1}\right)}^{2}}$, $Nu_{x}=\frac{xq_{w}}{k(T_{w}-T_{\infty })}$, $Sh_{x}=\frac{xq_{m}}{D_{B}\left(C_{w}-C_{\infty }\right)}$, $Nn_{x}=\frac{xq_{n}}{D_{m}n_{w}}$.

$\tau_{xz}=\mu \left[\partial_{z}u+\frac{Sb^{2}}{3}{\left(\partial_{z}u\right)}^{3}\right]$, $\tau_{yz}=\mu \left[\partial_{z}v+\frac{Sb^{2}}{3}{\left(\partial_{z}v\right)}^{3}\right]$, $q_{w}=-K\left[\partial_{z}T\right]$, $q_{m}=-D_{B}\left[\partial_{z}C\right]$, $q_{n}=-D_{m}\left[\partial_{z}n\right]$

Thus, we have the following:

$C_{fx}{\left(Re_{x}\right)}^{1/2}=-2\left[f^{\prime \prime }\left(0\right)+\frac{S}{3}ReDef^{\prime \prime 3}\left(0\right)\right]$, $C_{fy}{\left(Re_{x}\right)}^{1/2}=-2\left[g^{\prime }\left(0\right)+\frac{S}{3}ReDeg^{\prime 3}\left(0\right)\right]$,

$Nu_{x}{\left(Re_{x}\right)}^{-1/2}=-\theta^{\prime }\left(0\right)$, $Sh_{x}{\left(Re_{x}\right)}^{-1/2}=-\phi^{\prime }\left(0\right)$, $Nn_{x}{\left(Re_{x}\right)}^{-1/2}=-\chi^{\prime }\left(0\right)$.

where $\left(Re_{x}\right)=\frac{x^{2}\Omega sin\alpha }{\nu (1-st)}$ is the local Reynolds number.

## 3. Numerical Scheme

In this section, there are incorporated numerical outcomes from the nonlinearly accompanying ordinary differential Equations (1)–(6) with boundary conditions in Equation (Equation 8), which are combined utilizing the RK-4 technique. To carry out this analytical strategy, governing Equations (1)–(6) are combined into a first-order approach by introducing a distinct variable, as shown below:



$y_{1}^{\prime }=y_{2}$





$y_{2}^{\prime }=y_{3}$





$y_{3}^{\prime }=\frac{-1}{\left[1-\frac{S}{8}ReDef^{\prime \prime 2}\right]}\left[\frac{1}{2}f^{\prime 2}-ff^{\prime \prime }-2g^{2}-2A\left(f^{\prime }+\frac{1}{2}\eta f^{\prime \prime }\right)-Mf^{\prime }-2\lambda \left(\theta -Nr\phi -Rb\chi \right)\right]$





$y_{4}^{\prime }=y_{5}$





$y_{5}^{\prime }=\frac{-1}{\left[1-\frac{S}{2}ReDeg^{\prime 2}\right]}\left[-2fg^{\prime }+f^{\prime }g-2A\left(g+\frac{1}{2}\eta g^{\prime }\right)-Mg\right]$





$y_{6}^{\prime }=y_{7}$





$y_{7}^{\prime }=Pr\left[f\theta^{\prime }-\frac{1}{2}f^{\prime }\theta +A\left(2\theta +\frac{1}{2}\eta \theta^{\prime }\right)\right]-Nb\theta^{\prime }\phi^{\prime }-Nt\theta^{\prime 2}$





$y_{8}^{\prime }=y_{9}$





$y_{9}^{\prime }=Sc\left[f\phi^{\prime }-\frac{1}{2}f^{\prime }\phi +A\left(2\phi +\frac{1}{2}\eta \phi^{\prime }\right)\right]-\left(\frac{Nb}{Nt}\right)\theta^{\prime \prime }$





$y_{10}^{\prime }=y_{11}$





$y_{11}^{\prime }=Lb\left[f\chi^{\prime }-\frac{1}{2}f^{\prime }\chi +A\left(2\chi +\frac{1}{2}\eta \chi^{\prime }\right)\right]+Pe\left[\phi^{\prime \prime }+\chi^{\prime }\phi^{\prime }\right]$



along with the following boundary conditions.

f(0)=S, $f^{\prime }\left(0\right)=\Gamma_{u}f^{\prime \prime }\left(0\right)$, $-\Gamma_{v}g^{\prime }\left(0\right)+g\left(0\right)=1$, $-\Gamma_{T}\theta^{\prime }\left(0\right)+\theta \left(0\right)=1$,

$-\Gamma_{C}\phi^{\prime }\left(0\right)+\phi \left(0\right)=1$, $\chi \left(0\right)=1+\Gamma_{n}\chi^{\prime }\left(0\right)$, at η=0,

$f^{\prime }\left(\infty \right)\to 0$, g(∞)→0, θ(∞)→0, ϕ(∞)→0, χ(∞)→0, as η→∞.

## 4. Results and Discussion

For the validation of current outcomes, these are verified in the restricted cases when compared with preceding results (see Table 1). Table 1 provided the outcomes for heat transfer rate $-\theta^{\prime }\left(0\right)$, at cone walls for Pr, and λ when Nr=0.1,Sc=5 and all other parameters are zero. Among the current and previous findings, adequately sufficient accord is attained.

Graphical outcomes are obtained for different parameters when $\Gamma_{u}$ = 0.1, $\Gamma_{v}$ = 0.1, $\Gamma_{T}$ = 0.1, $\Gamma_{C}$ = 0.1, $\Gamma_{n}$ = 0.1, *M* = 1.0, *S* = 0.1, Pr = 6.8, Nb = 0.1, Nt = 0.1, Sc = 5.0, *A* = 0.1, Lb = 1.0, Pe = 0.1, λ = 0.1, Nr = 0.5, and Rb = 0.1. Moreover, the suitable ranges of parameters wwere increasing or decreasing behavior becomes smooth are taken as 0.05<A<0.2, 0.0<M<1.0, 0.1<Nr<0.3, 0.1<Rb<0.3, 0.1<λ<0.3, $0.0<\Gamma_{u}<1.0$, $0.0<\Gamma_{v}<1.0$, $0.0<\Gamma_{T}<1.0$, $0.0<\Gamma_{C}<1.0$, $0.0<\Gamma_{n}<1.0$, 2.0<Pr<4.0, 0.1<Nb<0.3, 0.01<Nt<0.1, 4.0<Sc<6.0, 0.1<Pe<0.3, and 3.0<Lb<5.0.

The plots in Figure 2, Figure 3, Figure 4, Figure 5, Figure 6 and Figure 7 delineated the distribution of velocities $f^{\prime }\left(\eta \right)$ and g(η) with a variation of leading parameters for two cases of nanoliquids and Sutterby nanofluids. It is revealed that the speed for Sutterby fluids is significantly slower than ordinary nanofluids. The larger viscous effects for Sutterby fluids impede the flow notably at the face as compared to that of nanofluids. The parabolic curve of $f^{\prime }\left(\eta \right)$ rises upward near the boundary due to the stretching cone and then it declines to far-off boundary conditions. Figure 2 portrays the slowing behavior of x-velocity $f^{\prime }\left(\eta \right)$ against mounting inputs of unsteadiness and magnetic parameters. In the presence of the magnetic force field, the reactive Lorentz force comes into play and retards the flow (see [31]). In unsteady flow, after the first jerk, the stretch in boundary diminishes and the fluid in the boundary slows down. It is observed that raising the unstable parameter drops the velocity distribution, which is associated by a decrease in the momentum boundary layer thickness in the profile, indicating that the unsteadiness factor declines the fluid velocity due to the spinning cone. From Figure 3, the slowing of fluid velocity is caused by the growing strength of Nr and Rb. The buoyancy effects put forth an adverse reaction to the flow in x-direction; hence, $f^{\prime }\left(\eta \right)$ decline. Vivid progress in x-velocity $f^{\prime }\left(\eta \right)$ is demonstrated in Figure 4 when mixed convection parameter λ and u-velocity slip parameter $\Gamma_{u}$ are improved. However, the higher inputs of v-slip velocity $\Gamma_{v}$ and thermal slip $\Gamma_{T}$ decrease the speed $f^{\prime }\left(\eta \right)$, as depicted in Figure 5. A meager decline in y-velocity g(η) is revealed against the unsteadiness seen in Figure 5. Moreover, Figure 6 and Figure 7 show the significant decrement of y-velocity g(η) when magnetic parameter *M*, u-velocity slip, and v-velocity slip parameters are made stronger. The sketch for temperature function θ(η) for nanofluids and Sutterby nanofluids is shown in Figure 8, Figure 9 and Figure 10. It is observed that temperature θ(η) for Sutterby nano-fluids is higher than that of nanofluids. In addition, Figure 8 indicate that temperature diminishes against the rising inputs of Prandlt number Pr and unsteadiness parameter *A*. This is due to the notion that enhancing unsteadiness improves heat loss due to the rotating cone, leading to a reduction in temperature distribution. Regardless of the reduction in the rate of heat transmission from the surface to the fluid for larger values of the unstable parameter, the cooling rate is significantly faster than the rate of cooling for the steady flow. However, the larger inputs of the parameter for Brownian motion Nb and thermophoresis Nt improves temperature distribution θ(η), as delineated in Figure 9. According to the physical nature of these two slip conditions, the rise in temperature is expected. The fast random motion of nano-particles in the base fluids (higher value of Nb) and the rapid movement of nanoentities from hotter to colder fluids (higher values of Nt) are responsible for raising temperature θ(η). Figure 10 expose that fluids temperature θ(η) is reduced against higher values of u-velocity slip $\Gamma_{u}$ and thermal velocity slip $\Gamma_{T}$, but it rises directly with v-velocity slip $\Gamma_{v}$. The normalized function ϕ(η) of nano-entities’ volume friction is mapped in Figure 11, Figure 12 and Figure 13. Volume friction ϕ(η) recedes against Sc, *A*, and Nb but it becomes enhanced with the increments in Nt, the thermophoretic parameter. Furthermore, u-velocity slip $\Gamma_{u}$ and solutal slip $\Gamma_{C}$ exert a receding impact on ϕ(η), whereas v-velocity slip $\Gamma_{v}$ enhances volume friction ϕ(η). The plots of micro-organism density χ(η) exhibit decrement in this function against the increment in Peclet number Pe and Lewis number Le, as shown in Figure 14. Similarly, Figure 15 displays a decreasing trend of χ(η) against u-velocity slip $\Gamma_{u}$ and unsteadiness parameter *A*. Figure 16 reveals that micro-organism density χ(η) increases with v-velocity slip parameter $\Gamma_{v}$, and it declines rapidly when motile density slip $\Gamma_{n}$ is improved. Table 2 and Table 3 in the respective order presents the enumeration of skin friction factors in x-direction and y-direction. Skin friction $-f^{\prime \prime }\left(0\right)$ along x-direction continues to decrease against parameter $\Gamma_{u}$*A*, *S*, *M*, Nr, and Rb, but it is enhanced directly with mixed convection parameter λ. Moreover, parameter *A*, *S*, and *M* incremented with $–f^{\prime \prime }\left(0\right)$ but parameter $\Gamma_{v}$ was reduced significantly. Table 4 enlists the local temperature rate on the cone surface to be reduced against $\Gamma_{T}$, Nt, and Nb but it increases with *A* and Pr. From Table 5, it is perceived that $–\phi^{\prime }\left(0\right)$ increases with *A*, Sc, Nb but it decreases against $\Gamma_{C}$ and Nb. Results for $–\chi^{\prime }\left(0\right)$ are registered in Table 6. Parameters *A*, Lb, and Pe enhance $–\chi^{\prime }\left(0\right)$, but it diminishes against $\Gamma_{n}$.

## 5. Conclusions

The objective and novelty of our manuscript is to explore the unsteady thermal and mass transportation of Sutterby nanofluids along enlarging cone surface where anisotropic boundary conditions are considered. In the presence of a magnetic field acting perpendicular to the axis of the cone’s bioconvection, thermal radiation and non-Fourier flux add to the physical aspects. The RK-4 technique and shooting strategy were used to combine numerical results from nonlinearly accompanying ordinary differential equations with boundary conditions. The impacts of distinct parameters, such as unsteadiness parameter, magnetic field parameter, and slip parameters, are portrayed graphically. Significant findings are stated below:In the x-direction, velocity $f^{\prime }\left(\eta \right)$ slows down against mounting inputs of *A*, *M*, Nr, Rb, $\Gamma_{v}$, and $\Gamma_{T}$ while it upsurges with λ and $\Gamma_{u}$.In the y-direction, velocity $f^{\prime }\left(\eta \right)$ slows down against mounting inputs of *A*, *M*, $\Gamma_{u}$, and $\Gamma_{v}$.A rising trend is observed in temperature profile θ(η) when Nb, Nt, and $\Gamma_{v}$ take larger values but it decreases when Pr, *A*, $\Gamma_{u}$, and $\Gamma_{T}$ are uplifted.Concentration profile ϕ(η) decrease when Sc, *A*, Nb, $\Gamma_{u}$, and $\Gamma_{c}$ intensifies but the opposite behavior is observed for Nt and $\Gamma_{T}$.Motile density profile χ(η) decreases when Pe, *A*, Lb, $\Gamma_{u}$, and $\Gamma_{n}$ intensifies but the opposite behavior is observed for $\Gamma_{v}$.The skin friction factor $-f^{\prime \prime }\left(0\right)$ along x-direction continues to decrease against $\Gamma_{u}$*A*, *S*, *M*, Nr, and Rb but it increases directly with mixed convection parameter λ. Moreover, parameters *A*, *S*, and *M* incremented $-f^{\prime \prime }\left(0\right)$ but parameter $\Gamma_{v}$ was reduced significantly.The local temperature rate on the cone surface decreased against $\Gamma_{T}$, Nt, and Nb but its increased with *A* and Pr.$-\phi^{\prime }\left(0\right)$ increases directly with *A*, Sc, and Nb, but it reduces against $\Gamma_{C}$ and Nb.Motile density number $-\chi^{\prime }\left(0\right)$ is directly enhanced with *A*, Lb, and Pe but it diminishes against $\Gamma_{n}$.

## 6. Future Directions

This problem can be extended as hybrid nanofluids by using finite element and finite difference schemes.

## Figures and Tables

**Figure 1 nanomaterials-12-02902-f001:**
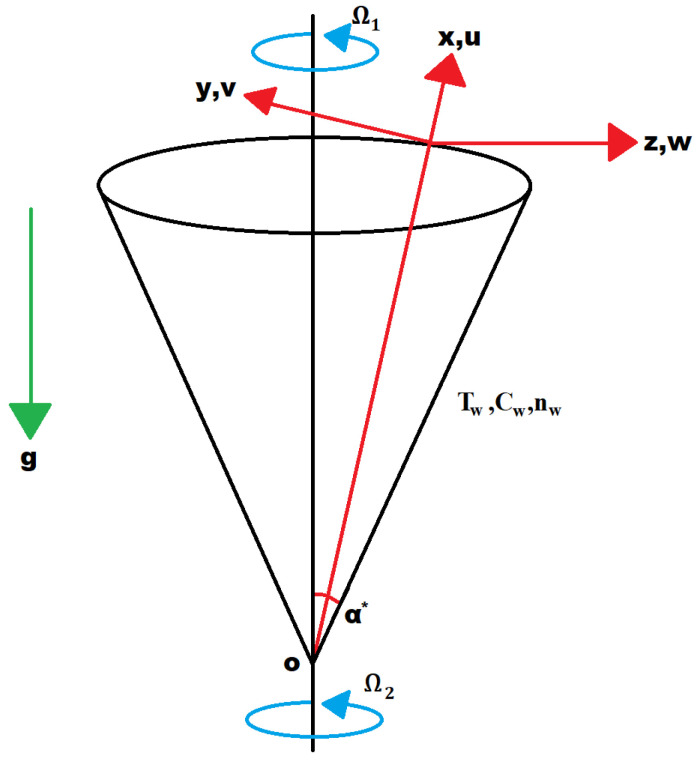
Flowchart.

**Figure 2 nanomaterials-12-02902-f002:**
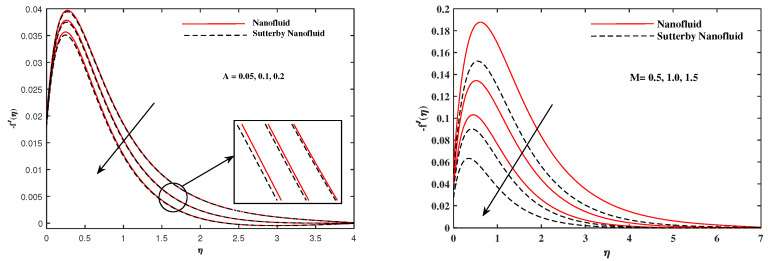
$f^{\prime }\left(\eta \right)$ fluctuate in x-direction with *A* and *M*.

**Figure 3 nanomaterials-12-02902-f003:**
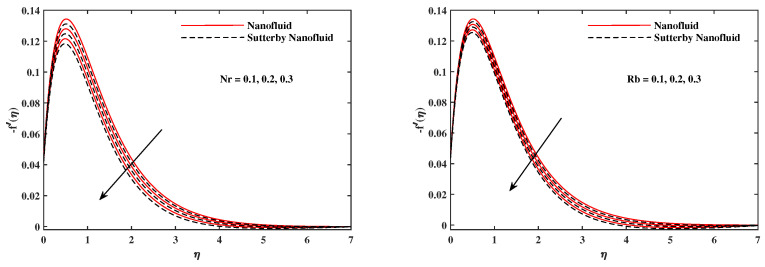
$f^{\prime }\left(\eta \right)$ fluctuate in x-direction with Nr and Rb.

**Figure 4 nanomaterials-12-02902-f004:**
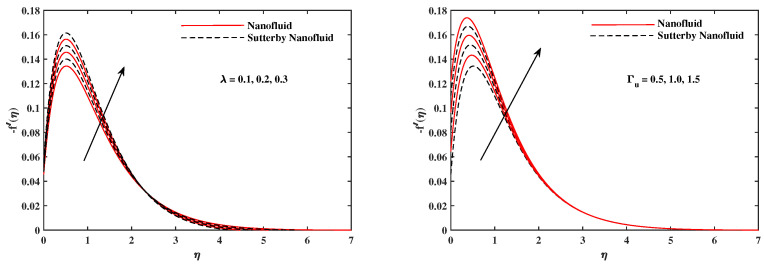
$f^{\prime }\left(\eta \right)$ fluctuate in x-direction with λ and $\Gamma_{u}$.

**Figure 5 nanomaterials-12-02902-f005:**
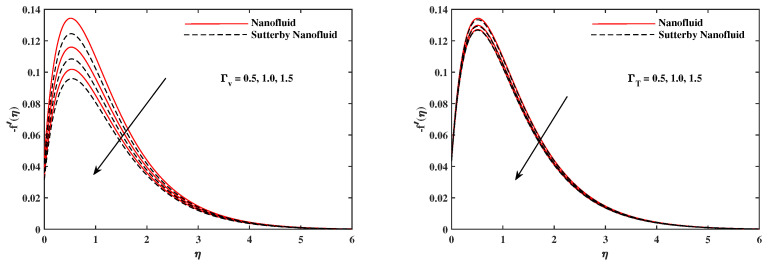
$f^{\prime }\left(\eta \right)$ fluctuate in x-direction with $\Gamma_{v}$ and $\Gamma_{T}$.

**Figure 6 nanomaterials-12-02902-f006:**
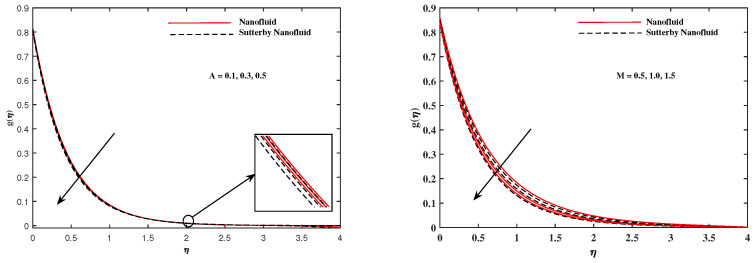
g(η) fluctuate in y-direction with *A* and *M*.

**Figure 7 nanomaterials-12-02902-f007:**
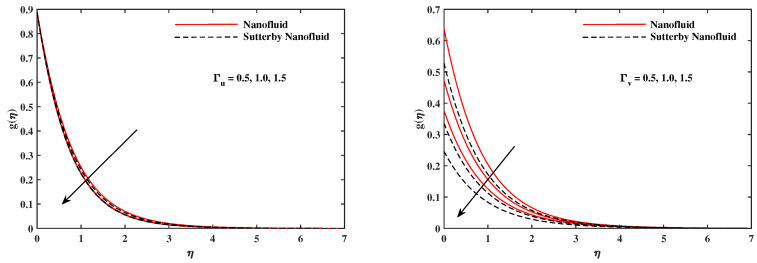
g(η) fluctuate in y-direction with $\Gamma_{u}$ and $\Gamma_{v}$.

**Figure 8 nanomaterials-12-02902-f008:**
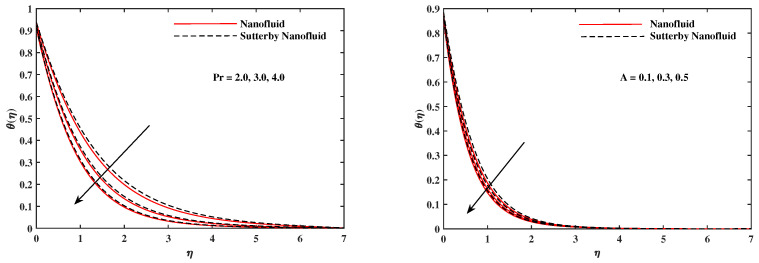
θ(η) fluctuate with Pr and *A*.

**Figure 9 nanomaterials-12-02902-f009:**
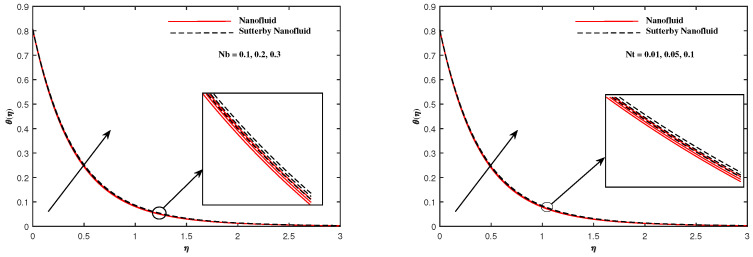
θ(η) with Nb and Nt.

**Figure 10 nanomaterials-12-02902-f010:**
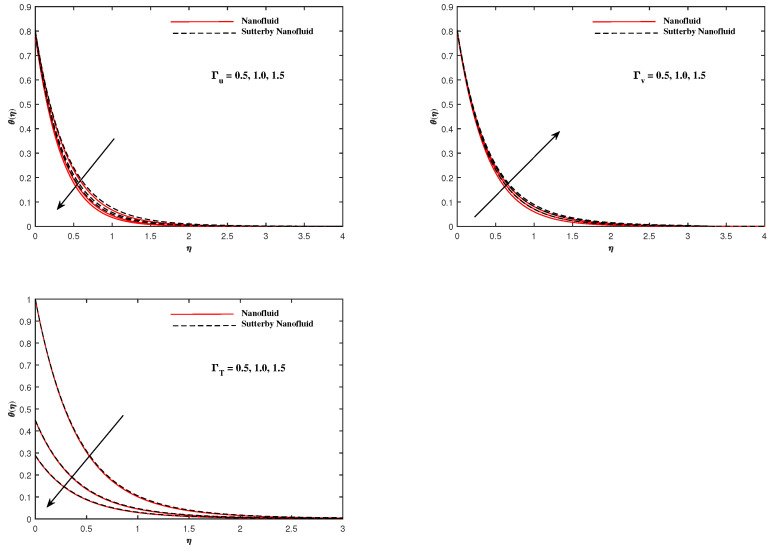
θ(η) fluctuate with $\Gamma_{u}$, $\Gamma_{v}$, and $\Gamma_{T}$.

**Figure 11 nanomaterials-12-02902-f011:**
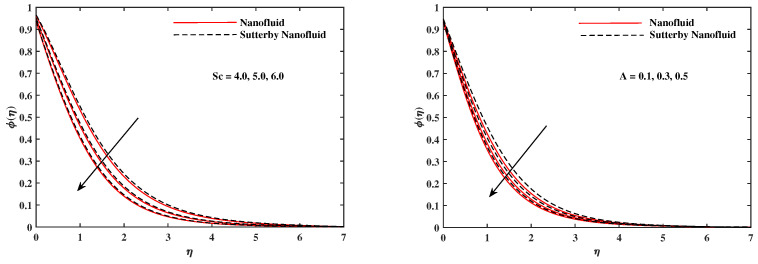
ϕ(η) fluctuate with Sc and *A*.

**Figure 12 nanomaterials-12-02902-f012:**
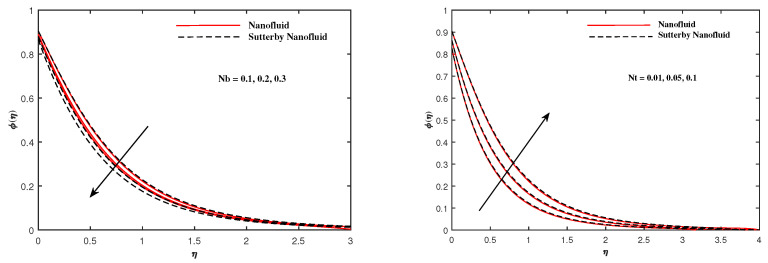
ϕ(η) fluctuate with Nb and Nt.

**Figure 13 nanomaterials-12-02902-f013:**
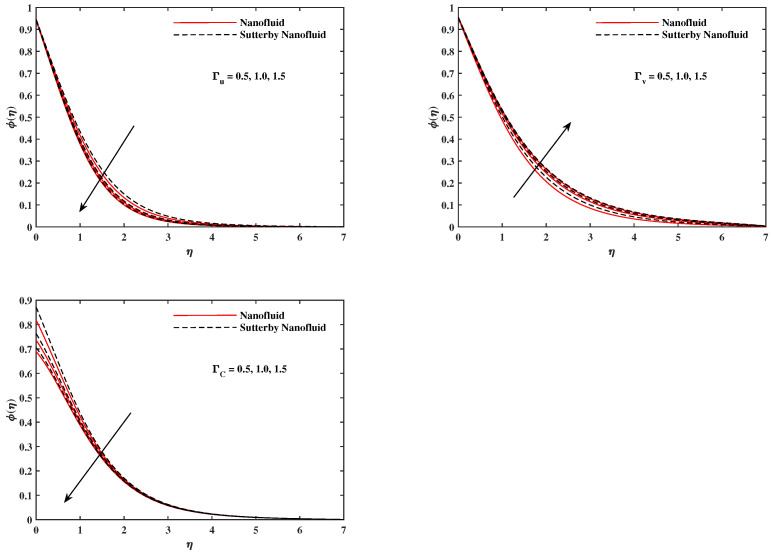
ϕ(η) fluctuate with $\Gamma_{u}$, $\Gamma_{v}$, and $\Gamma_{C}$.

**Figure 14 nanomaterials-12-02902-f014:**
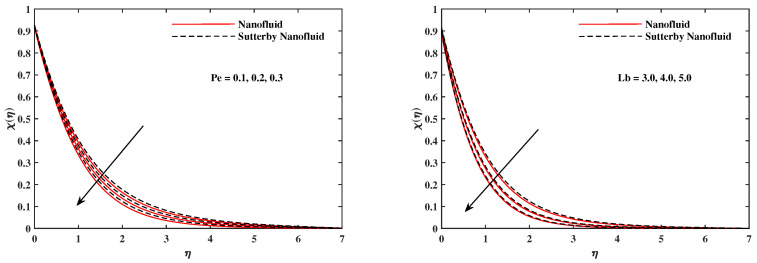
χ(η) fluctuate with Pe and Lb.

**Figure 15 nanomaterials-12-02902-f015:**
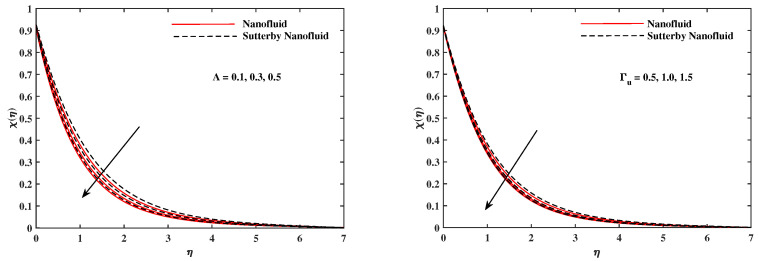
χ(η) fluctuate with *A* and $\Gamma_{u}$.

**Figure 16 nanomaterials-12-02902-f016:**
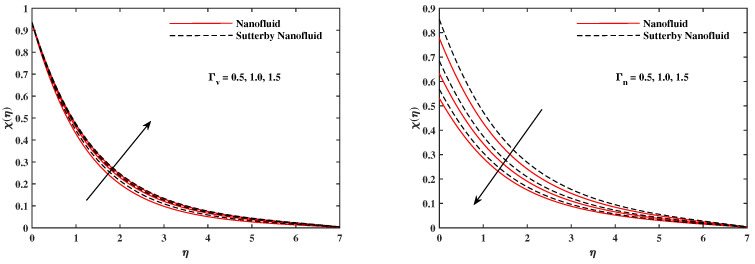
χ(η) fluctuate with $\Gamma_{v}$ and $\Gamma_{n}$.

**Table 1 nanomaterials-12-02902-t001:** For Pr and λ, comparative outputs of $-\theta^{\prime }\left(0\right)$.

Pr	λ	Anilkumar et al. [34]	Latiff et al. [31]	Present Results
0.7	0.0	0.4305	0.4287	0.4305
1.0	1.0	0.6127	0.6120	0.6082
0.7	0.0	0.5181	0.5180	0.5181
1.0	1.0	0.7005	0.7005	0.6955

**Table 2 nanomaterials-12-02902-t002:** Results for $-f^{\prime \prime }\left(0\right)$ in x-direction.

$\Gamma_{u}$	*A*	*S*	*M*	λ	Nr	Rb	$-f^{\prime \prime }\left(0\right)$
0.1	0.5	0.1	1.0	0.1	0.5	0.1	0.7127
0.2							0.6301
0.3							0.5644
0.1	0.1						0.9117
	0.3						0.7997
	0.5						0.7127
	0.5	0.1					0.7127
		0.3					0.7126
		0.6					0.7125
		0.1	1.0				0.7127
			1.5				0.6164
			2.0				0.5446
			1.0	0.1			0.7127
				0.2			0.7611
				0.3			0.8092
				0.1	0.1		0.7127
					0.3		0.7017
					0.5		0.6907
					0.1	0.1	0.7127
						0.2	0.7008
						0.3	0.6888

**Table 3 nanomaterials-12-02902-t003:** Results for $-f^{\prime \prime }\left(0\right)$ in y-direction.

$\Gamma_{v}$	*A*	*S*	*M*	$-f^{\prime \prime }\left(0\right)$
0.1	0.5	0.1	1.0	2.4291
0.2				2.1544
0.3				1.9381
0.1	0.1			2.0893
	0.3			2.2631
	0.5			2.4291
	0.5	0.1		2.4291
		0.3		2.4322
		0.6		2.4370
		0.1	1.0	2.4291
			1.5	2.6745
			2.0	2.8951

**Table 4 nanomaterials-12-02902-t004:** Results for $-\theta^{\prime }\left(0\right)$.

$\Gamma_{T}$	*A*	Pr	Nb	Nt	$-\theta^{\prime }\left(0\right)$
0.1	0.5	6.8	0.1	0.1	1.9587
0.2					1.6406
0.3					1.4108
0.1	0.1				1.1593
	0.3				1.6299
	0.5				1.9587
	0.5	6.8			1.9587
		7.0			1.9820
		7.2			2.0048
		6.8	0.1		1.9587
			0.15		1.9405
			0.2		1.9224
			0.1	0.01	1.9848
				0.05	1.9731
				0.1	1.9587

**Table 5 nanomaterials-12-02902-t005:** Results for $-\phi^{\prime }\left(0\right)$.

$\Gamma_{C}$	*A*	Sc	Nb	Nt	$-\phi^{\prime }\left(0\right)$
0.1	0.5	5.0	0.1	0.1	0.9493
0.2					0.8033
0.3					0.6922
0.1	0.1				0.5215
	0.3				0.7663
	0.5				0.9493
	0.5	4.0			0.7269
		5.0			0.9493
		6.0			1.1391
		5.0	0.1		0.9493
			0.15		1.2388
			0.2		1.3835
			0.1	0.01	1.7039
				0.05	1.3643
				0.1	0.9493

**Table 6 nanomaterials-12-02902-t006:** Results for $-\chi^{\prime }\left(0\right)$.

$\Gamma_{n}$	*A*	Lb	Pe	$-\chi^{\prime }\left(0\right)$
0.1	0.5	1.0	0.1	0.9560
0.2				0.8756
0.3				0.8076
0.1	0.1			0.5517
	0.3			0.7850
	0.5			0.9560
	0.5	0.5		0.6988
		1.0		0.9560
		1.5		1.1307
		1.0	0.1	0.9560
			0.3	1.0769
			0.5	1.2335

## Data Availability

Not applicable.

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
