# Peer review of "On Time-Dependent Rheology of Sutterby Nanofluid Transport across a Rotating Cone with Anisotropic Slip Constraints and Bioconvection"

_nanomaterials, 2022, doi:10.3390/nano12172902_

Round 1

Reviewer 1 Report

I reviewed the paper entitled “On time dependent Rheology of Sutterby nanofluid transport across a rotating cone with anisotropic slip constraints and bioconvection” with an interest.

In this paper, exploration for thermal distribution in comparative unsteady dynamics of Sutterby nano-fluids and ordinary nano-fluids pertaining to enlarging surface of a rotating cone is presented. Bio-convection of gyrotactic micro-organisms, Cattaneo-Christov and thermal radiative fluxes with magnetic field are significant physical aspects of the study. Anisotropic constraints on the cone surface are taken in to account. The leading formulation is transmuted into ordinary differential formate via similarity functions. Five coupled equations with non-linearly terms are resolved numerically through utilization of Matlab code for Runga-Kutta procedure. The parameters of buoyancy ratio, porosity of medium and bio-convection Rayleigh number decrease x-direction velocity. The slip parameter retard the y-direction velocity. The temperature for Sutterby fluids is as hotter level but its velocity is vividly slower as compared to those of nano-fluids. Temperature profile improves directly with thermopherosis, v-velocity slip and random motion of nano-entities.

In general, this manuscript is interesting. The results are useful for researchers and engineers. Overall, the paper is well-structured. I would recommend this manuscript for publication after answers to the questions. I have some questions and recommendations.

  1. Try to highlight the novelty in the abstract, end of the introduction and conclusion. These parts are the main sections for showing the novelty of the work
  2. The literature review has to be expanded. In the literature review part, you should perform a potent literature review and scrutinize the most relevant and recent published papers in high-quality journal articles. The literature review is one of the main parts of a scientific paper to show your novelty, and alert the readers that you are aware of the performed research studies
  3. The conclusion should be revised, and the main results should be summarized as the bullet points at the end of the conclusion. You should follow this structure: 1. A brief description of what have you done and what is your novelty? 2. How did you investigate the system and what are the main useful parameters? 3. Provide the primary results as the bullet points? 4. State the main limitations of this study and present some suggestions for future researches.

Author Response

Author’s response to Reviewer 1

The authors are highly thankful to the learned reviewer for his valuable time and suggestions. Our response to each of the comment is given below:

Reviewer comment 1: Try to highlight the novelty in the abstract, end of the introduction and conclusion. These parts are the main sections for showing the novelty of the work.

Author’s response: Agreed, novelty is highlighted in abstract section as well as in last paragraph of introduction section.

Reviewer comment 2: The literature review has to be expanded. In the literature review part, you should perform a potent literature review and scrutinize the most relevant and recent published papers in high-quality journal articles. The literature review is one of the main parts of a scientific paper to show your novelty, and alert the readers that you are aware of the performed research studies.

Author’s response: Literature review is expanded as per given instruction.

Reviewer comment 1: The conclusion should be revised, and the main results should be summarized as the bullet points at the end of the conclusion. You should follow this structure: 1. A brief description of what have you done and what is your novelty? 2. How did you investigate the system and what are the main useful parameters? 3. Provide the primary results as the bullet points? 4. State the main limitations of this study and present some suggestions for future researches.

Author’s response: This suggestion is followed.

Reviewer 2 Report

The reviewer thinks that this manuscript is really well presented, and the numerical solution here obtained could be very useful for researchers that deal with similar applications. The authors should answer to the following points in order to make the paper eligible for publication

  • Since many symbols are involved, it is suggested to use a nomenclature here
  • Scaling should be better explained here. For instance, how did the authors scale variables related to thermophoresis, i. e. the thermophoresis parameter? Which is its physical meaning?
  • Did the authors have the opportunity to compare their results with experimental data from the literature, if available?
  • Which is the meaning of the straight lines in Fig. 2 (left)? Please improve graphics everywhere in the manuscript to clarify the meaning of this lines
  • In Figs. 4 and 5, maxima of the curves are always shifting to the right, while this doesn't happen in other figures like Figs. 2 and 3. Please explain this aspect from a physical point of view
  • The introduction should be improved in order to clarify potential applications of the present study with references to biological sciences. For instance, it has been shown through the years that thermophoresis would have an impact on low-density lipoprotein deposition if some temperature gradients are applied [1], or for instance on drug delivery [2]. Therefore, the authors are invited to mention all this in order to improve their state of art

[1] Iasiello, M., Vafai, K., Andreozzi, A., & Bianco, N. (2019). Hypo-and hyperthermia effects on LDL deposition in a curved artery. Computational Thermal Sciences: An International Journal, 11(1-2).

[2] Vazifehshenas, F. H., & Bahadori, F. (2019). Investigation of Soret effect on drug delivery in a tumor without necrotic core. Journal of the Taiwan Institute of Chemical Engineers, 102, 17-24.

Author Response

Author’s response to Reviewer 2

The authors are highly thankful to the learned reviewer for his valuable time and suggestions. Our response to each of the comment is given below:

Reviewer comment 1: Since many symbols are involved, it is suggested to use a nomenclature here.

Author’s response: A nomenclature is added after the conclusion section.

Reviewer comment 2: Scaling should be better explained here. For instance, how did the authors scale variables related to thermophoresis, i. e. the thermophoresis parameter? Which is its physical meaning?

Author’s response: Agreed, The terms used to describe the nanoparticle slip factors are better scaled in Burginio model. Thermophoresis parameter Nt is proportional to theromophoretic coefficient DB and ratio of the heat capacitances. It is dimension less parameter. Similarly other parameters are scaled as described in the text.

Reviewer comment 3: Did the authors have the opportunity to compare their results with experimental data from the literature, if available?

Author’s response: We appreciate your observation and valuable suggestion to validity results with experimental data. Actually, we are expert in numerical computation and we are interested to compare numerical results with experimental data, but still we are facing much difficulty. In literature, many experts compare numerical results for limiting cases. For numerical validity, we compare our results with already publish paper for limiting cases (please see table 1).

Reviewer comment 4: Which is the meaning of the straight lines in Fig. 2 (left)? Please improve graphics everywhere in the manuscript to clarify the meaning of this lines.

Author’s response: Straight lines are the magnifying part of the figs which is made clear now. Rendering are made as per journal instructions.

Reviewer comment 5: In Figs. 4 and 5, maxima of the curves are always shifting to the right, while this doesn't happen in other figures like Figs. 2 and 3. Please explain this aspect from a physical point of view.

Author’s response: Author’s response: It is to mention that in the Fig.4 and Fig.5, the variation in the non dimensional velocity f’ is noticed due to the increments in velocity slip parameters gamma u and gammav(Гu and Гv). The higher inputs of these numbers improve the maxima of the velocity curve.  Physically, this situation occurs due   slip factors.   

Reviewer comment 6: The introduction should be improved in order to clarify potential applications of the present study with references to biological sciences. For instance, it has been shown through the years that thermophoresis would have an impact on low-density lipoprotein deposition if some temperature gradients are applied [1], or for instance on drug delivery [2]. Therefore, the authors are invited to mention all this in order to improve their state of art

[1] Iasiello, M., Vafai, K., Andreozzi, A., & Bianco, N. (2019). Hypo-and hyperthermia effects on LDL deposition in a curved artery. Computational Thermal Sciences: An International Journal, 11(1-2).

[2] Vazifehshenas, F. H., & Bahadori, F. (2019). Investigation of Soret effect on drug delivery in a tumor without necrotic core. Journal of the Taiwan Institute of Chemical Engineers, 102, 17-24.

Author’s response: This suggestion is followed.
